# Quantitative Analysis of Brain CT Perfusion in Healthy Beagle Dogs: A Pilot Study

**DOI:** 10.3390/vetsci10070469

**Published:** 2023-07-18

**Authors:** Soyon An, Gunha Hwang, Seul Ah Noh, Hee Chun Lee, Tae Sung Hwang

**Affiliations:** 1Institute of Animal Medicine, College of Veterinary Medicine, Gyeongsang National University, Jinju 52828, Republic of Korea; rapunzel9367@gmail.com (S.A.); hgh3634@gmail.com (G.H.); 2AniCom Medical Center, Animal Hospital, Seoul 04599, Republic of Korea; shtmfdk@gmail.com

**Keywords:** perfusion image, perfusion index, cerebrovascular circulation, multidetector computed tomography, dogs

## Abstract

**Simple Summary:**

Cerebral perfusion provides important information about capillary-level hemodynamics of the brain parenchyma, especially in conditions such as an ischemic stroke, brain tumors, and inflammation. The assessment of brain perfusion in dogs has been performed using computed tomography (CT), magnetic resonance (MR), sonography, single-photon emission computed tomography (SPECT), and 2-fluoro-2-deoxy-D-glucose positron emission tomography. However, these methods, except for CT, have limitations and difficulties in obtaining quantitative values. In addition, some of them are cost-prohibitive, particularly SPECT and MR imaging. CT scanners are widely available in clinical hospitals, making CT-based perfusion techniques more accessible for studies. The purpose of this study was to investigate the normal range of perfusion determined via CT in the brains of healthy dogs and to compare values between white matter and gray matter, differences in aging, and each hemisphere. The results of our study showed that the cerebral blood volume and cerebral blood flow values of white matter in the frontal, temporal, parietal, and occipital lobes were significantly lower than gray matter values. No significant differences were observed in the age-related variations or between the left and right hemispheres in other brain regions. In addition, quantitative values of CT perfusion parameters in a specific brain region were presented. This information provided by our study might contribute to a better understanding of brain perfusion in healthy dogs, potentially leading to the earlier diagnosis and improved treatment of brain diseases.

**Abstract:**

Brain computed tomography (CT) perfusion is a technique that allows for the fast evaluation of cerebral hemodynamics. However, quantitative studies of brain CT perfusion in veterinary medicine are lacking. The purpose of this study was to investigate the normal range of perfusion determined via CT in brains of healthy dogs and to compare values between white matter and gray matter, differences in aging, and each hemisphere. Nine intact male beagle dogs were prospectively examined using dynamic CT scanning and post-processing for brain perfusion. Regional cerebral blood volume (rCBV), regional cerebral blood flow (rCBF), mean transit time, and time to peak were calculated. Tissue ROIs were drawn in the gray matter and white matter of the frontal, temporal, parietal, and occipital lobes; caudate nucleus; thalamus; piriform lobe; hippocampus; and cerebellum. Significant differences were observed between the white matter regions and gray matter regions for rCBV and rCBF (*p* < 0.05). However, no significant differences were identified between hemispheres and between young and old groups in brain regions. The findings obtained in this study involving healthy beagle dogs might serve as a reference for regional CT perfusion values in specific brain regions. These results may aid in the characterization of various brain diseases in dogs.

## 1. Introduction

Cerebral perfusion provides important information about the capillary-level hemodynamics of the brain parenchyma, especially in conditions such as an ischemic stroke [1]. Brain tissue flow can be calculated using several parameters, including cerebral blood flow (CBF), cerebral blood volume (CBV), mean transit time (MTT), and time to peak (TTP). CBV refers to the total volume of flowing blood in a given volume of brain tissue, measured in mL/100 g [1]. CBF measures the volume of blood flow through a given brain volume per unit time, measured in mL/100 g/min [1]. MTT represents the average transit time of blood flow through a specific brain region. It is measured in seconds [1]. TTP is defined as the time required to reach peak enhancement according to the time-attenuation curve (TAC) [2,3].

Computed tomography perfusion (CTP) complements non-enhanced computed tomography (NECT) and computed tomography angiography (CTA) in the evaluation of brain parenchyma and other neurovascular disorders [4]. The first application of CTP for human brain imaging was proposed in 1980 [5]. However, due to limitations of CT acquisition and post-processing systems at the time, this method was not readily available [1,6]. Presently, routine CT protocols for imaging ischemic stroke in humans involve a NECT scan of the brain; supra-aortic CTA; and a dynamic, time-resolved CTP scan [7]. In dogs, the first attempt to measure perfusion using CT was made in beagle dogs in 1991 [8]. The assessment of brain perfusion in dogs has also been performed using magnetic resonance (MR) perfusion-weighted imaging [9,10,11], sonography [12], single-photon emission computed tomography (SPECT) [13,14], and 2-[18F]fluoro-2-deoxy-D-glucose positron emission tomography (PET) [15]. However, there are limitations and challenges associated with these methods in obtaining quantitative values. They can be less accessible, time-consuming, and cost-prohibitive, especially in the case of SPECT and MR imaging [6]. CT scanners are widely available in clinical hospitals, making CT-based perfusion techniques more accessible for studies [1,6]. However, quantitative studies on brain CT perfusion in veterinary medicine are lacking.

Aging increases cerebral arterial stiffness, which has a negative impact on cerebral blood flow regulation. It is an important risk factor affecting cardiovascular disease [16]. Various perfusion studies have reported differences in humans and dogs according to aging [13,15,17]. Brain perfusion and metabolism are associated with age-related global changes in humans [18] and dogs [19]. However, the reduction in regional frontal and temporal brain lobe perfusion with aging is consistent in human literature [18,20]. Moreover, regionally decreased metabolism has been demonstrated in aging dogs using 2-deoxy-D-glucose [15]. To the author’s knowledge, information regarding regional perfusion parameters for detailed anatomical brain regions in aging dogs using CT units is scarce.

We hypothesize that brain perfusion values might be decreased in older dogs just as in human patients. Thus, the objectives of this study were to determine normal ranges of regional perfusion parameters for individual brain regions and to investigate the differences in perfusion maps between young and old dogs. Data from this study could serve as the basis for future canine brain diseases.

## 2. Materials and Methods

### 2.1. Animals

The study population consisted of nine intact male beagle dogs, with a median weight of 10 kg (range: 7.9–12.5 kg) and a median age of 118 months (range: 32–118 months). These dogs had no evidence of cerebrovascular disease. Before this study, physical and neurologic examinations, blood tests, urinalysis, radiography, echocardiography, abdominal ultrasonography, and MR examinations were performed to ensure the absence of any remarkable findings and underlying diseases.

### 2.2. Preparation for CT

Anesthesia was induced using alfaxalone (2 mg/kg, IV, Alfaxan inj^®^, Careside. Co., Ltd., Seongnam, Korea) administered through an 18-gauge intravenous catheter with a 3-way stopcock in the cephalic vein. General anesthesia was maintained with isoflurane (Ifran^®^, Hana pharm. Co., Ltd., Seoul, Korea) in oxygen (2 L/min) via endotracheal intubation. Throughout these procedures, electrocardiography (ECG), oxygen saturation, and breathing were monitored. The heart rates of the dogs were monitored during scanning to ensure they remained above 100 beats/min. These rates were recorded before induction and after the CT scan.

All procedures were approved by the Institutional Animal Care and Use Committee at Gyeongsang National University. The dogs were cared for according to the Guidelines for Animal Experiments (GNU-220331-D0035) of Gyeongsang National University.

### 2.3. CT Image Acquisition

A 160-slice CT unit (Aquilion lightning 160, Canon Medical Systems, Otawara, Japan) was used for this study. The dogs were positioned in ventral recumbency on the CT table. Non-enhanced scans were initially performed from the nasal concha to the occipital bone, covering the entire brain. Scanning parameters for a non-enhanced helical CT were as follows: 120 kVp, 150 mAs, 512 × 512 matrix, 180 mm field of view (FOV), 0.75 s rotation time, 0.5 mm slice thickness, and 0.5 mm interval. Immediately after the non-contrast helical CT, a dynamic CT series was performed. The target section selected contained the lateral and largest cross section of the third ventricle, including carotid arteries [6]. The perfusion CT allowed for the assessment of four adjacent 10 mm thick slices. This choice was made to ensure a higher signal-to-noise ratio compared to acquiring eight adjacent 0.5 mm thick slices with the same acquisition parameters [17]. The dynamic volume CT scan took a total of 90 s to capture the entire passage of the contrast material bolus within the brain circulation [6]. Each study consisted of 29 sequential images with a 2 s scan interval. The scanning parameters for the dynamic volume CT were as follows: 80 kVp, 100 mAs effective tube current, 512 × 512 matrix, 150 mm FOV, 1 s rotation time, and 10 mm slice thickness.

The amount of contrast medium administered was determined based on the weight of each dog. Each dog was weighed before each contrast injection procedure to calculate the appropriate dosage of contrast medium. The contrast injection procedure was standardized for all experiments, with a total iodine dose of 370 mgI/kg administered at an injection rate of 2 mL/s. Subsequently, saline flushing at a rate of 2 mL/s using normal saline (0.9% NaCl) was performed using a dual injector (Salient^®^, Medrad Inc., Pittsburgh, PA, USA). Dynamic CT scanning was initiated 5 s prior to the injection of contrast media. This delay in the injection of the contrast material allowed for the acquisition of non-enhanced baseline images [6].

### 2.4. Perfusion Data Analysis

CT data were transferred to a workstation (Vitrea software; Vital Images Inc., Minnetonka, MN, USA) for computational analysis with an automated motion correction (in-plane) [21]. TACs were obtained by subtracting the regional mean baseline CT number in images obtained before the administration of the contrast material from the mean Hounds field unit (HU) values in serial contrast-enhanced images [6]. TACs were determined by manually placing circular regions of interest (ROIs) in the rostral cerebral artery (anterior cerebral artery in humans) and at the center of the dorsal sagittal venous sinus (superior sagittal venous sinus in humans) [17]. These data were filtered both spatially and temporally and automatically configured in the software.

The software utilized a Bayesian estimation algorithm [22] to calculate regional cerebral blood volume (rCBV, mL/100 g), regional cerebral blood flow (rCBF, mL/100 g/min), mean transit time (MTT, s), and time to peak (TTP, s) (Figure 1). A manual ROI was drawn using a mouse-guided cursor to outline free curved shapes within CT images. ROIs were defined to obtain tissue curves. A free curved shape of the ROI was adapted to best fit the shape of the structure being evaluated [17]. The following sets of tissue ROIs were selected: gray and white matter in frontal, temporal, parietal, and occipital lobes; bilateral caudate nucleus; thalamus; piriform lobe; hippocampus; and cerebellum [6,9,10,11,14,17] (Figure 2). Major blood vessels were excluded from the ROIs. The areas of ROIs were calculated in mm^2^.

### 2.5. Statistical Analysis

All statistical analyses were performed using a commercially available software package (SPSS 25.0, IBM, Armonk, NY, USA). The normal distribution of the data was confirmed using the Kolmogorov–Smirnov test. The Mann–Whitney U test was used to compare between young and old groups as well as between white matter and gray matter. Left–right differences within the two groups were evaluated using the Wilcoxon signed rank test. A *p*-value of less than 0.05 was considered statistically significant.

## 3. Results

A total of nine series of dynamic CT scans were conducted. The young-age group (*n* = 4) had a mean age of 32 months, and the old-age group (*n* = 5) had a mean age of 118 months, demonstrating a statistically significant difference (*p* = 0.016). Weight was also significantly (*p* = 0.016) different between the young group (mean 11.7 kg) and the old group (mean 9.1 kg). However, there was no significant difference in heart rate (*p* = 0.286) between the young group (mean 106.5 bpm) and the old group (mean 111.4 bpm). ECG and oxygen saturation remained stable throughout the experiments. Complications such as the paravasation of contrast material were not observed. The total mean values and standard deviations (SD) for each of the 13 brain regions are presented in Table 1.

### 3.1. Comparison of CT Perfusion Measurements between White Matter and Gray Matter

A total of four white matter ROIs and four gray matter ROIs were measured. All white matter regions exhibited significantly lower rCBV and rCBF values compared to gray matter regions. Specifically, in the frontal, temporal, parietal, and occipital lobes, the rCBV values in white matter were significantly lower than those in gray matter (*p* = 0.003, 0.003, <0.0001, and 0.001, respectively). Moreover, the rCBF values in the white matter of the frontal, temporal, parietal, and occipital lobes were significantly lower than those in gray matter (*p* ≤ 0.0001, 0.001, <0.0001, and 0.001, respectively). Differences between white matter and gray matter by *p* values are described in Table 2.

### 3.2. Comparison of CT Perfusion Measurements by Aging

There were no significant differences in the mean values for each of the 13 brain regions between the young and old groups, except for the HU value in the occipital gray matter (*p* = 0.034). In terms of the ROI size, the young group had significantly larger sizes in the frontal white matter (*p* = 0.021), frontal gray matter (*p* = 0.021), temporal gray matter (*p* = 0.001), parietal gray matter (*p* = 0.006), and occipital white matter (*p* = 0.034) than the old group. Details of ROIs, HU values, and regional perfusion values within each group are presented in Table 3.

### 3.3. Comparison of CT Perfusion Measurements by the Side of Hemisphere

There were no significant differences in the mean values for any of the 13 brain regions between right and left sides of the hemisphere. Details of ROIs, HU values, and regional perfusion values within each side of the hemisphere are shown in Table 4.

## 4. Discussion

The aim of this study was to establish reference values of perfusion determined using CTP in the brains of healthy beagle dogs and to compare values between white matter and gray matter, differences in aging, and each hemisphere. We evaluated various brain regions and compared the effects of aging in healthy beagle dogs. This study involved nine beagle dogs. Perfusion values were recorded based on detailed brain structures. The brain regions were selected according to the anatomy of their function and on the basis of CT images [6,9,10,11,14,17]. Furthermore, we compared values of CBV, CBF, MTT, and TTP between the young and old age groups.

Brain CTP provides essential information for patients with acute or chronic cerebrovascular diseases [1]. In humans, changes in brain perfusion patterns have been identified in various brain diseases such as ischemic stroke, brain tumors, and inflammation in the brain [23,24,25,26]. Therefore, understanding normal perfusion levels is crucial for diagnosing brain conditions. In human stroke cases, CBV ranges from 0 to 1.5 mL/100 g and CBF decreases from 0 to 10 mL/100 g/min in patients [23]. Another study has reported that CBF is reduced by 34% or more in the infarction region compared to that in a normal hemisphere [24]. Several studies have revealed that CBV, CBF, and permeability are elevated in brain tumors and have used these measurements to differentiate glioma grades [25,26]. In veterinary medicine, studies have revealed alterations in perfusion parameters in cerebrovascular diseases such as ventriculomegaly, epilepsy, and brain tumors using MR perfusion and CTP [9,11,27,28,29,30]. For instance, a study using MR has found that CBF in the periventricular white matter is significantly lower in dogs with ventriculomegaly [9]. Another study using MR has revealed lower CBF and longer TTP in dogs with idiopathic epilepsy than in healthy dogs, with the most pronounced effects observed in the piriform lobe, thalamus, and temporal gray matter [11]. Moreover, brain tumors in animals show higher CBF, CBV, and permeability than surrounding brain tissue, consistent with human studies [27,30]. In our study, general neurological and hematological examinations as well as MR imaging were performed to rule out subclinical diseases in dogs. Consequently, we established reference data for CBV, CBF, MTT, and TTP in various brain structures using dynamic CTP in beagle dogs. Conducting further studies on brain CTP comparing normal and diseased dogs might aid in the diagnosis of various brain diseases in canine patients.

A previous study has found that cerebral white matter has significantly lower CBV and CBF values than gray matter in both dogs and humans [6]. The results of the present study were consistent with this previous finding, showing that the values of white matter in frontal, temporal, parietal, and occipital lobes were significantly lower than those of gray matter.

Normal CBF values in humans vary from 20 mL/100 g/min in white matter to 80 mL/100 g/min in gray matter [31]. The CBF values observed in our study for dogs were much higher than the published CBF values for humans. CBF values in the brains of rats range from 105 ± 16 mL/100 g/min to 139 ± 19 mL/100 g/min [32], which are also greater than those in humans. A recent study using MR perfusion has revealed that the mean CBF value is 146.8 ± 27.3 mL/100 g/min in the semioval center and 257.9 ± 72.4 mL/100 g/min in the temporal cerebral cortex on dorsal perfusion-weighted images of dogs [10]. These findings suggest that CBF is generally higher in dogs than in humans. Another possible reason for the difference could be the use of isoflurane as an anesthetic. In contrast to human medicine, CT imaging techniques for the brains of dogs require anesthesia [10]. Studies in rats have shown that CBF increases when isoflurane is used as an anesthetic compared to pentobarbital or fentanyl. Such an increase is attributed to intracerebral vasodilation, particularly pronounced in the thalamus and subcortical areas [33,34]. Therefore, the influence of anesthesia cannot be excluded from our study. Thus, when comparing results with clinical measurements, the anesthesia method should be standardized among dogs.

Aging has been associated with changes in brain perfusion due to several reasons. Reduced synaptic density and neuronal size can lead to decreased metabolic activity, resulting in reduced perfusion with increasing age [35]. Increased arterial stiffness associated with advanced aging is also related to lower CBF [36]. In human studies, CBF has been found to decrease with age in various brain regions, including the cingulate, parahippocampus, superior temporal white matter, medial frontal white matter, posterior parietal white matter, and left insular cortex [35]. The decrease in CBF ranges from 3.9 mL/min to 4.8 mL/min (0.52%) per year [37]. Reduced CBF associated with aging has also been reported in dogs. In a single photon-emission CT study, dogs older than 96 months showed significantly decreased CBF values in the frontal lobe, temporal lobe, and subcortical nuclei compared to younger control dogs [13]. Another human study has reported a decrease in CBV in gray matter but an increase in MTT in both white and gray matter with increasing age [37]. In our study, we did not find significant differences in any perfusion measurements between young and old dogs. One possible explanation for this might be the small number of dogs in each group used in this study. Another explanation could be that the young-aged group was fully mature or that the old-aged group was not old enough to exhibit significant changes in brain perfusion. Previous studies have defined “senior” as the last 25% of the estimated lifespan until the end of life [38]. One study has reported that the regional cerebellar metabolic rate in beagle dogs over 14 years of age was significantly higher than that in the younger group [39]. These factors might have hindered our ability to detect significant differences. Alternatively, this finding can be interpreted as a result of an age-related study using CTP in healthy dogs.

A previous human study has found significantly greater brain perfusion parameters in the left cerebral hemisphere than in the right hemisphere, attributed to volume differences [20]. Similar findings have been observed in dogs using SPECT [40]. However, other studies using SPECT [14] and MR perfusion [10] did not identify significant differences in perfusion between the left and right cerebral hemispheres of dogs. In our study, the left and right hemispheres showed no significant differences in ROIs for any perfusion measurements.

The present study has several limitations. Firstly, the sample population was small, and only medium-sized male dogs were used. CBF is known to be higher in women than in men, potentially due to differences in hematocrits, blood viscosity, or circulating estrogen concentrations [37]. However, veterinary medicine studies have found no significant differences between male and female dogs [10]. To strengthen our findings, further studies involving a larger and more diverse sample of dogs in terms of size, age, sex, and breed are necessary. Secondly, we only utilized one method for dynamic CT imaging, which involved a total iodine dose of 370 mgI/kg at an injection rate of 2 mL/s with saline flushing. In a pilot study for this research, we found that high-speed contrast medium injection was effective. However, its clinical application was challenging. The high injection flow velocity can increase the pressure on peripheral veins in small dogs [41]. Therefore, we experimented with an optimal protocol that involved a suitably high but manageable injection speed. Thirdly, ROIs were drawn using only CT images, not MR images. If ROIs were not accurately delineated, especially in distinguishing gray matter and white matter, results may differ from expected outcomes, as observed in earlier studies [28]. Hence, it is crucial to carefully select an ROI and consider using high-quality MR images to ensure the accurate differentiation of variables between white matter and gray matter and to delineate detailed anatomy of the brain. Fourthly, we did not collect or compare results with invasive criterion-referenced techniques for direct perfusion measurements. However, the estimation of cerebrovascular variables using CT has been validated in rabbits [42] and dogs [6,28]. The values obtained in our study closely resembled those reported in dogs, rabbits, and humans [6,28,42]. Fifth, a slice thickness of 10 mm, in a medium-sized dog, is a large volume of the brain. If focal and minor perfusion changes were present, they were likely to remain undetected.

## 5. Conclusions

The results of this study demonstrated the use of CT to estimate rCBV, rCBF, MTT, and TTP values in 13 ROIs in the brain regions of nine healthy beagle dogs. Our measurements were consistent with those reported in human and animal studies. Information provided by our study might contribute to a better understanding of brain perfusion in healthy dogs, potentially leading to the earlier diagnosis and improved treatment of brain diseases. Moreover, further studies will need to be more useful and obtain a lot of data using CTP in dogs of more variable sizes and breeds.

## Figures and Tables

**Figure 1 vetsci-10-00469-f001:**
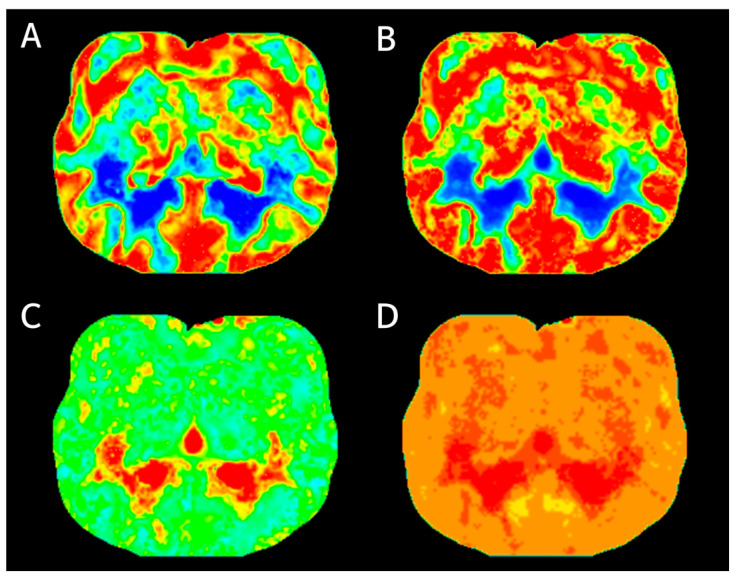
Perfusion maps of the transverse imaging at the thalamus region level were generated from identical source data using software. The perfusion maps include (**A**) regional cerebral blood volume, (**B**) regional cerebral blood flow, (**C**) mean transit time, and (**D**) time to peak.

**Figure 2 vetsci-10-00469-f002:**
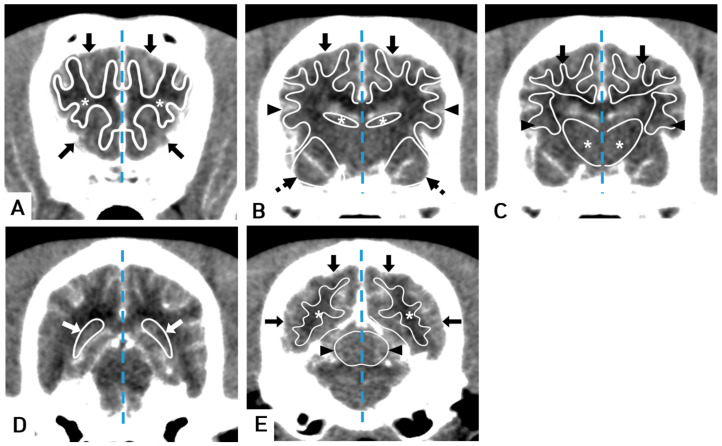
Transverse brain CT images showing the evaluated regions of interest (ROIs). (**A**) CT scan with superimposed illustrations showing ROIs drawn around the frontal white matter (asterisk) and frontal gray matter (arrow). (**B**) The parietal gray matter (arrow), temporal gray matter (arrowheads), caudate nucleus (asterisk), and piriform lobe (dotted arrow). (**C**) The parietal white matter (arrow), temporal white matter (arrowheads), and thalamus (asterisk). (**D**) The hippocampus (white arrow). (**E**) The occipital white matter (asterisk), occipital gray matter (arrow), and cerebellum (arrowheads). Note: Window level of 80 HU and width of 150 HU, brain midline (dotted line).

**Table 1 vetsci-10-00469-t001:** Mean values (±standard deviations) of CT measurements for each brain region (left and right region of interest values averaged; *n* = 18).

	ROI (mm²)	HU Value	rCBV (mL/100 g)	rCBF (mL/100 g/min)	MTT (s)	TTP (s)
Frontal white matter	1.59 ± 0.39	44.2 ± 6.95	9.63 ± 2.48	118.64 ± 27.46	4.35 ± 1.15	22.97 ± 5.85
Frontal gray matter	2.53 ± 0.56	61.93 ± 12.42	13.11 ± 3.42	142.27 ± 14.09	4.03 ± 0.71	22.33 ± 5.44
Temporal white matter	1.20 ± 0.21	43.62 ± 6.95	9.54 ± 2.57	117.53 ± 28.06	4.32 ± 1.11	22.98 ± 5.85
Temporal gray matter	1.72 ± 0.56	57.27 ± 10.05	12.94 ± 3.2	141.43 ± 14.13	4.11 ± 0.75	22.47 ± 5.46
Parietal white matter	0.91 ± 0.15	41.61 ± 6.69	8.67 ± 2.72	102.77 ± 34.23	5.27 ± 1.8	23.72 ± 6.38
Parietal gray matter	1.27 ± 0.19	63.38 ± 8.9	13.58 ± 3.21	141.93 ± 14.5	4.19 ± 0.73	22.44 ± 5.52
Occipital white matter	1.66 ± 0.32	44.8 ± 5.65	9.04 ± 2.71	108.84 ± 33.08	5.03 ±1.61	23.48 ± 6.18
Occipital gray matter	2.36 ± 0.72	66.51 ± 9.69	13.41 ± 3.64	140.46 ± 16.21	4.26 ± 0.8	22.31 ± 5.39
Caudate nucleus	0.28 ± 0.04	42.96 ± 11.78	10.72 ± 3.29	108.49 ± 22.25	6.15 ± 1.84	23.89 ± 5.97
Thalamus	0.68 ± 0.13	50.47 ± 8.32	12.98 ± 3.1	136.37 ± 18.61	4.46 ± 0.94	22.79 ± 5.63
Piriform lobe	1.63 ± 0.37	56.31 ± 7.35	12.48 ± 3.06	138.27 ± 17.83	4.11 ± 0.77	22.51 ± 5.52
Hippocampus	1.04 ± 0.15	55.87 ± 6.16	13.19 ± 2.49	139.44 ± 13.53	4.36 ± 0.77	22.7 ± 5.51
Cerebellum	1.81 ± 0.35	56.26 ± 6.1	13.22 ± 3.18	137.71 ± 20.82	4.37 ± 0.87	22.73 ± 5.66

**Table 2 vetsci-10-00469-t002:** Comparison of perfusion measurement for white versus gray matter of frontal, temporal, parietal, and occipital lobe (*p* value).

	Frontal White vs. Gray Matter	Temporal White vs. Gray Matter	Parietal White vs. Gray Matter	Occipital White vs. Gray Matter
rCBV (mL/100 g)	0.003	0.003	<0.0001	0.001
rCBF (mL/100 g/min)	<0.0001	0.001	<0.0001	0.001
MTT (s)	0.962	0.788	0.159	0.079
TTP (s)	0.506	0.506	0.547	0.447

**Table 3 vetsci-10-00469-t003:** Mean values (±standard deviations) of CT measurements for individual brain regions within each age group (left and right ROI values averaged; young group, *n* = 8; old group, *n* = 10).

		ROI (mm²)	HU value	rCBV (mL/100 g)	rCBF (mL/100 g/min)	MTT (s)	TTP (s)
Frontal white matter	Young	1.78 ± 0.33	46.53 ± 1.35	10.48 ± 2.64	120.96 ± 26.95	4.45 ± 1.09	24.26 ± 5.63
	Old	1.44 ± 0.39	42.34 ± 9.02	8.96 ± 2.25	116.79 ± 29.18	4.27 ± 1.25	21.93 ± 6.1
	*p* value	**0.021**	0.515	0.315	0.762	0.762	0.633
Frontal gray matter	Young	2.9 ± 0.51	64.49 ± 2.14	14.21 ± 3.37	144.45 ± 9.75	4.14 ± 0.58	23.45 ± 5.28
	Old	2.24 ± 0.42	59.89 ± 16.65	12.22 ± 3.36	140.53 ± 17.13	3.95 ± 0.83	21.43 ± 5.67
	*p* value	**0.021**	0.515	0.203	0.237	0.696	0.696
Temporal white matter	Young	1.28 ± 0.19	46.01 ± 1.96	10.25 ± 2.69	120.48 ± 27.93	4.38 ± 1.04	24.28 ± 5.64
	Old	1.14 ± 0.21	41.71 ± 8.9	8.97 ± 2.46	115.17 ± 29.44	4.28 ± 1.22	21.94 ± 6.09
	*p* value	0.122	0.315	0.573	0.696	0.762	0.762
Temporal gray matter	Young	2.18 ± 0.51	62.49 ± 4.37	14.0 ± 3.41	143.34 ± 9.97	4.24 ± 0.64	23.64 ± 5.26
	Old	1.36 ± 0.25	53.1 ± 11.51	12.1 ± 2.91	139.9 ± 17.15	4.01 ± 0.85	21.53 ± 5.71
	*p* value	**0.001**	0.203	0.237	0.762	0.633	0.696
Parietal white matter	Young	0.95 ± 0.18	44.28 ± 3.77	9.61 ± 3.08	108.34 ± 33.25	5.16 ± 1.48	24.81 ± 6.09
	Old	0.88 ± 0.12	39.48 ± 7.88	7.92 ± 2.28	98.32 ± 36.11	22.84 ± 6.79	22.84 ± 6.79
	*p* value	0.315	0.237	0.408	0.460	0.965	0.762
Parietal gray matter	Young	1.4 ± 0.13	66.61 ± 3.51	14.69 ± 3.31	144.3 ± 9.61	4.33 ± 0.7	23.65 ± 5.34
	Old	1.16 ± 0.16	60.79 ± 11.11	12.69 ± 2.99	140.03 ± 17.79	4.09 ± 0.77	21.48 ± 5.75
	*p* value	**0.006**	0.360	0.315	0.762	0.408	0.696
Occipital white matter	Young	1.83 ± 0.09	46.29 ± 1.43	10.11 ± 2.81	112.59 ± 29.72	5.18 ± 1.1	24.71 ± 5.78
	Old	1.52 ± 0.37	43.61 ± 7.43	8.19 ± 2.42	105.85 ± 36.84	4.92 ± 1.98	22.5 ± 6.62
	*p* value	**0.034**	0.515	0.203	0.897	1.000	0.762
Occipital gray matter	Young	2.48 ± 0.71	71.95 ± 5.41	15.06 ± 3.75	143.94 ± 9.78	4.54 ± 0.69	23.76 ± 5.46
	Old	2.26 ± 0.75	62.16 ± 10.36	12.08 ± 3.1	137.68 ± 20.06	4.04 ± 0.85	24.65 ± 5.7
	*p* value	0.173	**0.034**	0.173	0.360	0.173	0.633
Caudate nucleus	Young	0.3 ± 0.0	38.7 ± 6.85	9.61 ± 1.36	102.94 ± 22.76	6.4 ± 1.66	25.41 ± 5.4
	Old	0.26 ± 0.05	46.36 ± 14.02	11.61 ± 4.13	112.93 ± 21.98	5.95 ± 2.04	22.68 ± 6.4
	*p* value	0.173	0.315	0.146	0.573	0.573	0.633
Thalamus	Young	0.68 ± 0.16	50.79 ± 3.7	13.33 ± 3.37	136.64 ± 15.92	4.46 ± 0.65	23.98 ± 5.33
	Old	0.68 ± 0.1	50.22 ± 10.95	12.71 ± 3.02	136.16 ± 21.37	4.45 ± 1.15	21.84 ± 5.96
	*p* value	0.762	0.897	0.696	0.762	0.829	0.515
Piriform lobe	Young	1.58 ± 0.27	57.6 ± 2.3	13.24 ± 3.39	139.21 ± 15.93	4.23 ± 0.64	23.74 ± 5.35
	Old	1.68 ± 0.44	55.27 ± 9.76	11.88 ± 2.8	137.52 ± 20.04	4.02 ± 0.88	21.53 ± 5.74
	*p* value	0.515	0.696	0.408	0.897	0.829	0.633
Hippocampus	Young	1.1 ± 0.13	55.3 ± 4.48	13.7 ± 2.23	140.66 ± 9.35	4.5 ± 0.75	24.0 ± 5.16
	Old	1.0 ± 0.15	56.33 ± 7.45	12.79 ± 2.65	138.47 ± 16.59	4.25 ± 0.82	21.66 ± 5.83
	*p* value	0.237	0.573	0.633	0.633	0.573	0.762
Cerebellum	Young	1.83 ± 0.21	56.35 ± 1.65	14.16 ± 3.43	139.51 ± 18.9	4.43 ± 0.76	23.98 ± 5.47
	Old	1.8 ± 0.44	56.19 ± 8.26	12.46 ± 2.92	136.26 ± 23.15	4.33 ± 0.99	21.73 ± 5.9
	*p* value	0.633	0.965	0.315	0.633	0.762	0.762

**Table 4 vetsci-10-00469-t004:** Mean values (±standard deviations) of CT measurements for individual brain regions within each side of hemisphere (*n* = 9).

		ROI (mm²)	HU value	rCBV (mL/100 g)	rCBF (mL/100 g/min)	MTT (s)	TTP (s)
Frontal white matter	Right	1.59 ± 0.4	44.49 ± 7.04	9.61 ± 2.45	119.03 ± 27.7	4.36 ± 1.17	22.96 ± 5.98
	Left	1.59 ± 0.4	43.91 ± 7.29	9.66 ± 2.65	118.26 ± 28.9	4.34 ± 1.2	22.98 ± 6.07
	*p* value	1.000	0.931	0.931	1.000	1.000	1.000
Frontal gray matter	Right	2.53 ± 0.58	62.37 ± 12.69	13.06 ± 3.42	142.41 ± 14.42	4.03 ± 0.71	22.38 ± 5.59
	Left	2.53 ± 0.58	61.5 ± 12.89	13.16 ± 3.63	142.13 ± 14.62	4.03 ± 0.76	22.28 ± 5.62
	*p* value	1.000	0.931	0.863	0.863	1.000	0.931
Temporal white matter	Right	1.2 ± 0.21	43.74 ± 7.2	9.6 ± 2.81	117.14 ± 29.8	4.38 ± 1.16	22.98 ± 6.05
	Left	1.2 ± 0.21	43.5 ± 7.13	9.48 ± 2.49	117.91 ± 28.01	4.27 ± 1.13	22.98 ± 6.0
	*p* value	1.000	1.000	1.000	1.000	0.796	1.000
Temporal gray matter	Right	1.72 ± 0.58	57.21 ± 10.24	12.9 ± 3.4	140.43 ± 14.73	4.17 ± 0.8	22.48 ± 5.67
	Left	1.72 ± 0.58	57.33 ± 10.48	12.99 ± 3.18	142.42 ± 14.32	4.06 ± 0.74	22.46 ± 5.58
	*p* value	1.000	1.000	1.000	0.666	0.489	0.931
Parietal white matter	Right	0.91 ± 0.15	41.98 ± 7.25	8.92 ± 3.0	102.9 ± 34.87	5.33 ± 1.79	23.7 ± 6.59
	Left	0.91 ± 0.15	41.24 ± 6.49	8.42 ± 2.58	102.64 ± 35.69	5.21 ± 1.92	23.73 ± 6.57
	*p* value	1.000	0.730	0.605	0.863	1.000	1.000
Parietal gray matter	Right	1.27 ± 0.19	63.29 ± 9.43	13.64 ± 3.33	142.02 ± 14.04	4.21 ± 0.75	22.44 ± 5.7
	Left	1.27 ± 0.19	63.47 ± 8.92	13.51 ± 3.28	141.83 ± 15.8	4.18 ± 0.74	22.44 ± 5.69
	*p* value	1.000	0.796	0.796	0.863	0.863	1.000
Occipital white matter	Right	1.66 ± 0.33	44.73 ± 5.44	9.2 ± 2.63	108.8 ± 32.41	5.04 ± 1.58	23.44 ± 6.34
	Left	1.66 ± 0.33	44.87 ± 6.18	8.89 ± 2.93	108.89 ± 35.71	5.02 ± 1.73	23.52 ± 6.4
	*p* value	1.000	0.863	0.730	1.000	1.000	1.000
Occipital gray matter	Right	2.36 ± 0.74	65.93 ± 10.6	13.24 ± 3.85	139.66 ± 16.6	4.3 ± 0.85	22.56 ± 5.77
	Left	2.36 ± 0.74	67.09 ± 9.29	13.57 ± 3.63	141.27 ± 16.77	4.22 ± 0.81	22.06 ± 5.32
	*p* value	1.000	0.666	0.931	0.730	0.666	0.730
Caudate nucleus	Right	0.28 ± 0.04	42.56 ± 13.22	10.51 ± 3.52	105.72 ± 24.55	6.29 ± 1.98	23.93 ± 6.21
	Left	0.28 ± 0.04	43.36 ± 10.93	10.93 ± 3.24	111.26 ± 20.78	6.01 ± 1.8	23.86 ± 6.1
	*p* value	1.000	0.796	0.863	0.546	0.863	0.863
Thalamus	Right	0.68 ± 0.13	49.81 ± 8.27	12.66 ± 3.12	135.21 ± 18.3	4.4 ± 0.77	22.74 ± 5.73
	Left	0.68 ± 0.13	51.13 ± 8.81	13.31 ± 3.24	137.53 ± 19.94	4.51 ± 1.12	22.83 ± 5.87
	*p* value	1.000	0.796	0.666	0.387	1.000	0.796
Piriform lobe	Right	1.7 ± 0.36	56.18 ± 7.97	12.37 ± 3.18	137.29 ± 19.92	4.12 ± 0.81	22.53 ± 5.74
	Left	1.7 ± 0.36	56.43 ± 7.16	12.6 ± 3.12	139.26 ± 16.63	4.1 ± 0.77	22.49 ± 5.64
	*p* value	0.489	0.863	0.796	0.931	0.863	0.931
Hippocampus	Right	1.04 ± 0.15	55.27 ± 6.36	12.91 ± 2.44	137.07 ± 14.29	4.43 ± 0.86	22.79 ± 5.69
	Left	1.04 ± 0.15	56.48 ± 6.27	13.48 ± 2.65	141.82 ± 13.12	4.29 ± 0.72	22.61 ± 5.67
	*p* value	1.000	0.666	0.666	0.340	0.666	0.730
Cerebellum	Right	1.81 ± 0.36	55.88 ± 6.26	13.19 ± 3.26	136.97 ± 23.08	4.39 ± 0.92	22.74 ± 5.86
	Left	1.81 ± 0.36	56.64 ± 6.3	13.24 ± 3.3	138.44 ± 19.67	4.36 ± 0.87	22.71 ± 5.82
	*p* value	1.000	0.666	0.931	0.931	0.931	1.000

## Data Availability

Not applicable.

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
