# Peer review of "Quantitative Analysis of Brain CT Perfusion in Healthy Beagle Dogs: A Pilot Study"

_vetsci, 2023, doi:10.3390/vetsci10070469_

Round 1
Reviewer 1 Report
The paper aims to address the limited availability of quantitative studies on brain CT perfusion in veterinary medicine, aiming to bridge this research gap. While there have been previous investigations on brain CT perfusion in humans and other veterinary studies utilizing imaging modalities such as MRI, this paper focuses on the initial reporting of CT perfusion parameters in multiple brain regions. These findings establish baseline values and offer valuable insights for future disease-related investigations.
The paper has clear organization and fluent language, demonstrating its well-written nature. The primary limitation of the study is the relatively small and homogenous sample population. But this limitation is well summarized in the discussion section. The paper aligns well with the scope of the journal, and I would recommend its consideration with some minor modifications.
Specific comments (MA = major, MI = minor, OP = optional)
1. [MA] Title: I believe the title does not adequately encompass all the findings presented in the paper. From my understanding, the results can be divided into two main parts: 1. Normal CT perfusion parameters in multiple brain regions 2. Group comparison: a) white versus gray matter b) old vs young c) left vs right hemisphere. The current title only mentions parts 1 and 2b, while omitting 2a and 2c.
2. [MI] “The aim of this study was to gather information about normal perfusion parameters in 21 healthy dogs.”
It does not cover the whole paper. I would suggest including similar sentences from the abstract for a more comprehensive coverage. “The purpose of this study was to investigate the normal range of perfusion determined via CT in brains of healthy dogs and to compare values between white matter and gray matter, differences of aging, and each hemisphere.”
3. [MA] Simple Summary: “The results of our study showed that cerebral blood volume and cerebral blood flow values of white matter in the frontal, temporal, parietal and occipital lobes were significantly lower than gray matter values. In addition, quantitative values of CT perfusion parameters in a specific brain region were presented.”
Even though no significant differences were observed among the age and hemisphere groups, it is important to report these findings in order to provide a comprehensive analysis of the data.
4. [MI] Abstract: P1, L33. “Nine beagle dogs” -> "Nine male beagle dogs" (add "male")
5. [MA] Introduction: P2 L 51-53. Add references to both CBV and CBF as you did for MTT and TTP.
6. [OP] Introduction: P2 L 51-53. “It is measured in mL/100g.” To enhance conciseness, you could combine it with the previous sentence.
For example, CBV refers to the total volume of flowing blood in a given volume of brain tissue, measured in mL/100g.
7. [MA] Introduction: P2 L 68. Please briefly discuss the limitations and difficulties in obtaining quantitative values and provide references.
8. [MA] 2.1 Animals: P2 L 89. Please report weight and age (mean, std, range).
9. [OP] P2 L94. I recommend incorporating a new subsection labeled ‘2.2. Preparation for CT’.
10. [MA] P3 L 138. “These data were filtered both spatially and temporally.” Please briefly discuss the filtering function you used.
11. [MA] Figure 1: I did not find E) in the figure.
12. [MA] Figure 2: “around the frontal white matter (asterisk) and frontal white matter (arrow).” I assume one is frontal gray matter.
13. [MI] P5 L 168. “A p-value of less than 0.05 was considered statistically significant.” Please provide a reference for 0.05 or discuss how you selected this value.
14. [MI] P5 L170. “A total of 9 series of dynamic CT scans were conducted. Their mean weight was 10.3 kg, with a mean age of 79.8 months.” I believe this belongs to the ‘2.1 Animals’ subsection. I also recommend reporting standard deviation and range values.
15. [OP] P6 L 185 - 196. I recommend condensing the paragraph by avoiding the repetition of similar sentences for each region. This will help streamline the content and make it more concise.
16. [OP] 5. Conclusions P11 L 316. Any future plan to solve the limitations you discussed in the discussion section?
Author Response
Reviewer 1
The paper aims to address the limited availability of quantitative studies on brain CT perfusion in veterinary medicine, aiming to bridge this research gap. While there have been previous investigations on brain CT perfusion in humans and other veterinary studies utilizing imaging modalities such as MRI, this paper focuses on the initial reporting of CT perfusion parameters in multiple brain regions. These findings establish baseline values and offer valuable insights for future disease-related investigations.
The paper has clear organization and fluent language, demonstrating its well-written nature. The primary limitation of the study is the relatively small and homogenous sample population. But this limitation is well summarized in the discussion section. The paper aligns well with the scope of the journal, and I would recommend its consideration with some minor modifications.
â–º We sincerely thank you for your efforts in reviewing our manuscript. We have included our point-by-point responses to your comments below.
Specific comments (MA = major, MI = minor, OP = optional)
- [MA] Title: I believe the title does not adequately encompass all the findings presented in the paper. From my understanding, the results can be divided into two main parts: 1. Normal CT perfusion parameters in multiple brain regions 2. Group comparison: a) white versus gray matter b) old vs young c) left vs right hemisphere. The current title only mentions parts 1 and 2b, while omitting 2a and 2c.
â–º We appreciate your comment. We think it’s a very good point and agree with your opinion. However, there was an opinion that it was unreasonable to evaluate the effect of age and setting a normal value with the number of 9 animals, so the title was modified to pilot study. Please let me know if you have a better opinion on the title.
We change the title of this paper “Quantitative Analysis of Brain CT Perfusion in Healthy Beagle Dogs: A pilot study”
an Experimental Investigation of Feasibility
- [MI] “The aim of this study was to gather information about normal perfusion parameters in 21 healthy dogs.”
It does not cover the whole paper. I would suggest including similar sentences from the abstract for a more comprehensive coverage. “The purpose of this study was to investigate the normal range of perfusion determined via CT in brains of healthy dogs and to compare values between white matter and gray matter, differences of aging, and each hemisphere.”
â–º Thank you for your comment. As your suggestion, we change the sentences more clearly to cover the whole paper. Please find the revised Simple summary and Discussion below:
(P1 L21) Simple summary: From “The aim of this study was to gather information about normal perfusion parameters in healthy dogs. We evaluated various brain regions and compared effects of aging in healthy beagle dogs.” to “The purpose of this study was to investigate the normal range of perfusion determined via CT in brains of healthy dogs and to compare values between white matter and gray matter, differences of aging, and each hemisphere.”
Discussion: From “The aim of this study was to gather information about normal perfusion parameters in healthy dogs and assess the influence of aging using dynamic CTP.” to “The aim of this study was to establish reference values of perfusion determined using CTP in brains of healthy beagle dogs and to compare values between white matter and gray matter, differences of aging, and each hemisphere.”
- [MA] Simple Summary: “The results of our study showed that cerebral blood volume and cerebral blood flow values of white matter in the frontal, temporal, parietal and occipital lobes were significantly lower than gray matter values. In addition, quantitative values of CT perfusion parameters in a specific brain region were presented.”
Even though no significant differences were observed among the age and hemisphere groups, it is important to report these findings in order to provide a comprehensive analysis of the data.
â–º (P1 L25-26) We appreciate your comment. As your suggestion, we add sentence on the simple summary section. Please find the added Simple Summary below:
“No significant differences were observed in the age-related variations or between the left and right hemispheres in other brain regions.”
- [MI] Abstract: P1, L33. “Nine beagle dogs” -> "Nine male beagle dogs" (add "male")
â–º (P1 L34) Thank you for your comment. As you said, we modify from “Nine beagle dogs” to "Nine intact male beagle dogs"
- [MA] Introduction: P2 L 51-53. Add references to both CBV and CBF as you did for MTT and TTP.
â–º (P2 L53-54)We appreciate your comment. As you said, we add references for CBV and CBF.
[1] Konstas, A.A.; Goldmakher, G. V.; Lee, T.Y.; Lev, M.H. Theoretic basis and technical implementations of CT perfusion in acute ischemic stroke, part 1: Theoretic basis. AJNR. Am. J. Neuroradiol. 2009
- [OP] Introduction: P2 L 51-53. “It is measured in mL/100g.” To enhance conciseness, you could combine it with the previous sentence.
For example, CBV refers to the total volume of flowing blood in a given volume of brain tissue, measured in mL/100g.
â–º (P2 L52-54) Thank you for your comment. As you said, we combine these two sentences on the Introduction section.
“CBV refers to the total volume of flowing blood in a given volume of brain tissue, measured in mL/100g [1]. CBF measures the volume of blood flow through a given brain volume per unit time, measured in mL/100g/min [1].”
- [MA] Introduction: P2 L 68. Please briefly discuss the limitations and difficulties in obtaining quantitative values and provide references.
â–º (P2 69-71) We appreciate your comment. Following the comments you gave, we add the word about limitation and difficulties shortly. Please find the revised Introduction below:
“However, there are limitations and challenges associated with these methods in ob-taining quantitative values. They can be less accessible, time-consuming, and cost-prohibitive, especially in the case of SPECT and MR imag.”
- [MA] 2.1 Animals: P2 L 89. Please report weight and age (mean, std, range).
â–º (P2, L91-93) Thank you for your comment. As you said, we add STD and range of weight and age on the Animals of Materials and Methods section.
“The study population consisted of nine male beagle dogs, with a median weight of 10 kg (range: 7.9 – 12.5 kg) and a median age of 118 months (range: 32 – 118 months).”
- [OP] P2 L94. I recommend incorporating a new subsection labeled ‘2.2. Preparation for CT’.
â–º (P3 L97) We appreciate your comment. As you recommend, we incorporate the new subsection for preparation for CT on the Materials and Methods section.
“2.2. Preparation for CT”
- [MA] P3 L 138. “These data were filtered both spatially and temporally.” Please briefly discuss the filtering function you used.
â–º (P3 L144) Thank you for your comment. There was no filter that we could choose from in the experiment process, and it is a method that is automatically configured in the software. As your suggestion, we add some words about filtering. Please find the revised Perfusion data analysis below:
“These data were filtered both spatially and temporally and automatically configured in the software.”
- [MA] Figure 1: I did not find E) in the figure.
â–º (P4 L 156-158) Thank you for your comment. We apologize for your confusion, and we modified it to the sentence below;
“The perfusion maps include (A) regional cerebral blood volume, (B) regional cerebral blood flow, (C) mean transit time, and (D) time to peak.”
- [MA] Figure 2: “around the frontal white matter (asterisk) and frontal white matter (arrow).” I assume one is frontal gray matter.
â–º (P5 L 162) Thank you for your comment. We apologize for the misused expression and thank you for the opportunity to correct the wrong word. As you said, we change the word “frontal white matter” to “frontal gray matter”.
- [MI] P5 L 168. “A p-value of less than 0.05 was considered statistically significant.” Please provide a reference for 0.05 or discuss how you selected this value.
â–º Thank you for your comment. We evaluated the significant difference using both the Mann-Whitney U test and Wilcoxon signed-rank test, with a p-value of less than 0.05, by comparing only the gray matter and white matter, as well as the young and old, and left and right sides in each region. Each evaluation was conducted to determine the significant difference between two variables. We considered a p-value less than 0.05 as indicating a significant difference.
- [MI] P5 L170. “A total of 9 series of dynamic CT scans were conducted. Their mean weight was 10.3 kg, with a mean age of 79.8 months.” I believe this belongs to the ‘2.1 Animals’ subsection. I also recommend reporting standard deviation and range values.
â–º (P2, L91-93) Thank you for your comment. We apologize for the missed expression of the animal’s information and appreciate you for the opportunity to correct. As you said, we move the sentences and add STD and range of weight and age on the Animals of Materials and Methods section.
“The study population consisted of nine male beagle dogs, with a median weight of 10 kg (range: 7.9 – 12.5 kg) and a median age of 118 months (range: 32 – 118 months).”
- [OP] P6 L 185 - 196. I recommend condensing the paragraph by avoiding the repetition of similar sentences for each region. This will help streamline the content and make it more concise.
â–º (P6 L188-194) We appreciate your comment. As you recommend, we condense the paragraph on the Results. Please find the revised Results below:
“All white matter regions exhibited significantly lower rCBV and rCBF values compared to gray matter regions. Specifically, in the frontal, temporal, parietal, and occipital lobes, rCBV values in white matter were significantly lower than those in gray matter (P = 0.003, 0.003, <0.0001, 0.001, respectively). Moreover, rCBF values in the white matter of the frontal, temporal, parietal, and occipital lobes were significantly lower than those in gray matter (P = <0.0001, 0.001, <0.0001, 0.001, respectively).”
- [OP] 5. Conclusions P11 L 316. Any future plan to solve the limitations you discussed in the discussion section?
â–º (P11 L 322-324) Thank you for your comment. As you suggest, we add the sentence for future plan to solve the limitations. Please find the revised Conclusions below:
“Moreover, further study will need to more useful as to obtain a lot of data using CTP in dogs of more variable sizes and breeds in dogs.”

Reviewer 2 Report
Comments to the authors of the manuscript `Quantitative Analysis of Brain CT Perfusion in Healthy Beagle Dogs: Establishing Normal Values and Investigating Effect of Age`
Normal values and effect of age cannot be established with 9 subjects. The title should be rephrased.
Line 89: intact dogs?
Line 126: with a power injector?
Line 144: From the images I guess the ROI were drawn free handed (not polygonal)
Line 151: transverse imaging of the brain at which level?
Line 155: Tranverse brain images
Figure 2: some of the black arrows are not well visible in the images, maybe white or red will ease the reading
Table 3 and 4: it would help the reader if the statistical significance could be marked (like in bold or with an asterisk)
Line 157: frontal gray matter
Line 171: because of the small number of dogs, body weight and age should be reported as median and range.
Line 218: The aim of the study should be rephrased as well. With this study design, you can obtain reference or baseline values.
Line 221: please rephrase: Do you mean that the brain regions were selected accordingly to the anatomy?
Line 229: …decreases from…to
Line 242: established reference data or baseline data
Line 250: Can you provide explanation for that?
Line 286: Similarly, in the group of `young` dogs, dogs were also actually already mature.
Among the limitations the slice thickness should also be mentioned:
Slice thickness of 10mm, in a middle sized dog, is large volume of the brain. If focal and minor perfusion changes were present, they are likely to remain undetected.
Minor editing of the language is recommended.
Author Response
Reviewer 2
Comments to the authors of the manuscript `Quantitative Analysis of Brain CT Perfusion in Healthy Beagle Dogs: Establishing Normal Values and Investigating Effect of Age`
Normal values and effect of age cannot be established with 9 subjects. The title should be rephrased.
â–º We appreciate your comment. We think it’s a very good point and agree with your opinion. A pilot study was added in accordance with the opinion that it was unreasonable to evaluate the effect of age and setting a normal value with the number of 9 animals. Please let me know if you have a better opinion on the title.
“Quantitative Analysis of Brain CT Perfusion in Healthy Beagle Dogs: A pilot study”
Line 89: intact dogs?
â–º (P2 L91) We appreciate your comment. All dogs used in this study were intact males. Therefore, an intact male was added.
Line 126: with a power injector?
â–º (P3 L131) Thank you for your comment. As you said, we add the words for method of inject saline. Please find the revised the CT image acquisition below:
“saline flushing at a rate of 2 mL/sec using normal saline (0.9% NaCl) was performed using dual injector (Salient®, Medrad Inc., Pittsburgh, PA, USA).”
Line 144: From the images I guess the ROI were drawn free handed (not polygonal)
â–º (P4 L148,149) We appreciate your comment. As you commend, we change the word from “polygonal shape” to “free curved” on 2.4. perfusion data analysis.
Line 151: transverse imaging of the brain at which level?
â–º (P4 L 156) Thank you for your comment. We apologize for your confusion, and we add the expression of the level of perfusion image. Please find the revised the title of Figure 1. below:
“Perfusion maps of the transverse imaging at the thalamus region level were generated from identical source data using software.”
Line 155: Tranverse brain images
â–º (P4 L 160) We appreciate your comment. As you said, we change the word from “Axial” to “Transverse” on the Figure 2.
Figure 2: some of the black arrows are not well visible in the images, maybe white or red will ease the reading
â–º (Figure 2D and L164) Thank you for your comment. As you recommend, we change the color of arrows in Figure 2. In Figure D, the black arrow was not clearly visible, so it was modified with a white arrow.
Table 3 and 4: it would help the reader if the statistical significance could be marked (like in bold or with an asterisk)
â–º (Table 3 and 4) We appreciate your comment. Thank you for recommendation and we modify for statistical significant with bold text on the Table 3 and 4.
Line 157: frontal gray matter
â–º (P5 L 162) Thank you for your comment. We apologize for the misused expression and thank you for the opportunity to correct the wrong word. As you said, we change the word “frontal white matter” to “frontal gray matter”.
Line 171: because of the small number of dogs, body weight and age should be reported as median and range.
â–º (P2, L91-93) We appreciate your comment. As you said, we add the range of dog’s weight and age, and move on the Materials and Methods section.
“The study population consisted of nine male beagle dogs, with a median weight of 10 kg (range: 7.9 – 12.5 kg) and a median age of 118 months (range: 32 – 118 months).”
Line 218: The aim of the study should be rephrased as well. With this study design, you can obtain reference or baseline values.
â–º (P9 L 215-217) Thank you for your comment. As your suggestion, we modify the sentence of the aim of this study. Please find the revised Discussion below:
From “The aim of this study was to gather information about normal perfusion parameters in healthy dogs and assess the influence of aging using dynamic CTP.” to “The aim of this study was to establish reference values of perfusion determined using CTP in brains of healthy beagle dogs and to compare values between white matter and gray matter, differences of aging, and each hemisphere.”
Line 221: please rephrase: Do you mean that the brain regions were selected accordingly to the anatomy?
â–º (P9 L 219-221) We appreciate your comment. We apologize for the misused expression of the word for selection brain regions. As you said, we rephrase the sentences of brain region. Please find the revised Discussion below:
From “The selection of brain regions was guided by their different functions and the basis of CT images [6,9–11,14,17].” to “The brain regions were selected accordingly to the anatomy of their function and the basis of CT images [6,9–11,14,17].”
Line 229: …decreases from…to
â–º (P10 L 228) Thank you for your comment. We apologize for making mistake and thank you for the opportunity to correct the error. We change “CBF decreases to 0 to 10 mL/100g/min in patients” to CBF decreases from 0 to 10 mL/100g in patients” on the Discussion section.
Line 242: established reference data or baseline data
â–º (P10 L 242) We appreciate your comment. As you said, we modify the word from “we established normal parameters for CBV, CBF, MTT and TTP” to “we established reference data for CBV, CBF, MTT and TTP” on the Discussion section.
Line 250: Can you provide explanation for that?
â–º Thank you for your comment. The reason is that the cortex, which contains most gray matter, is folding with a lot of multiple wrinkles. Therefore, gray matter has a larger volume than it looks. Here is a reference paper about this theory:
Mota, Bruno, et al. "White matter volume and white/gray matter ratio in mammalian species as a consequence of the universal scaling of cortical folding." Proceedings of the National Academy of Sciences 116.30 (2019): 15253-15261.
Line 286: Similarly, in the group of `young` dogs, dogs were also actually already mature.
â–º (P11 L 292) We appreciate your comment. We think it’s a very good point and agree with your opinion. As you said, we add the sentence for dog of young group were fully mature. Please find the revised Discussion below:
From “Another explanation could be that the old-aged group was not old enough to exhibit significant changes in brain perfusion.” to “Another explanation could be that the young-aged group was fully mature or old-aged group was not old enough”
Among the limitations the slice thickness should also be mentioned:
Slice thickness of 10 mm, in a middle sized dog, is large volume of the brain. If focal and minor perfusion changes were present, they are likely to remain undetected.
â–º (P12 L 327-328) Thank you for your comment. As your recommendation, we add sentences about limitation of slice thickness on the last paragraph of Discussion section.
“Fifth, slice thickness of 10 mm, in a middle sized dog, is large volume of the brain. If focal and minor perfusion changes were present, they are likely to remain undetected.”
